# The Association of Central Sensitisation with Depression, Anxiety, and Somatic Symptoms: A Cross-Sectional Study of a Mental Health Outpatient Clinic in Japan

**DOI:** 10.3390/life14050612

**Published:** 2024-05-10

**Authors:** Takeaki Takeuchi, Kazuaki Hashimoto, Akiko Koyama, Keiko Asakura, Masahiro Hashizume

**Affiliations:** 1Department of Psychosomatic Medicine, School of Medicine, Toho University, Tokyo 143-8541, Japan; kazuaki.hashimoto@med.toho-u.ac.jp (K.H.); akiko.02.koyama@med.toho-u.ac.jp (A.K.); hashi2@med.toho-u.ac.jp (M.H.); 2Department of Environmental and Occupational Health, School of Medicine, Toho University, Tokyo 143-8541, Japan; keiko.asakura@med.toho-u.ac.jp

**Keywords:** anxiety, central sensitisation, depression, somatic symptoms

## Abstract

For patients with chronic pain and persistent physical symptoms, understanding the mechanism of central sensitisation may help in understanding how symptoms persist. This cross-sectional study investigated the association of central sensitisation with depression, anxiety, and somatic symptoms. Four hundred and fifteen adults attending an outpatient psychosomatic clinic were evaluated. Participants completed the Hospital Anxiety and Depression Scale, Somatic Symptom Scale 8, and the Central Sensitisation Inventory. The relationships between these factors were examined using descriptive statistics and multiple logistic regression analyses. The mean age was 42.3 years, and 59% were female. The disorders included adjustment disorders (n = 70), anxiety disorders (n = 63), depressive disorders (n = 103), feeding and eating disorders (n = 30), sleep–wake disorders (n = 37), somatic symptoms and related disorders (n = 84), and others (n = 28). In multiple logistic regression analyses, higher central sensitisation was associated with more severe anxiety, depression, and somatic symptoms after controlling for potential confounders. In the disease-specific analysis, somatic symptoms correlated more positively with central sensitisation than with depression or anxiety. Central sensitisation and depression, anxiety, and somatic symptoms were associated with patients attending an outpatient clinic. These findings highlight the importance of evaluating depression, anxiety, and somatic symptoms when assessing central sensitisation.

## 1. Introduction

Central sensitisation (CS) refers to the increased sensitivity of the central nervous system to stimuli, which can lead to chronic pain, fatigue, and other symptoms; it is defined as an increased response of the central nervous system [1,2,3]. As concrete phenomena, ‘hyperalgesia’, or markedly increased sensitivity to painful stimuli, and ‘allodynia’, or experiencing pain in response to non-noxious stimuli, are known [4]. CS has been primarily studied in relation to neurological and musculoskeletal disorders, such as rheumatoid arthritis, persistent neck pain, and fibromyalgia, which are often present in chronic pain syndromes [5]. There have been multiple reports on the relationship between CS and pain [6,7,8,9,10]. However, research on the relationship between CS and psychiatric symptoms is limited. Among the limited research, depression and anxiety, which are the most frequent psychiatric symptoms, have been discussed in several previous studies. In patients with fibromyalgia, CS is positively associated with depression and anxiety [11,12,13]. In breast cancer survivors, CS may positively correlate with depression and anxiety, although this difference was not statistically significant [14]. In painful temporomandibular disorders, CS is not associated with depression but is positively associated with anxiety [15]. In chronic lower back pain, high trait anxiety-related personality characteristics predict the extent of CS symptoms [16]. However, some studies have found negative relationships between CS and depression and anxiety [17,18,19]. The relationship between CS and depression and anxiety has been inconsistent in previous studies, and the study population has been limited to specific pain-related diseases.

Although studies have reported that CS, depression, and anxiety are related to pain [20,21,22], few studies have directly examined these relationships in patients with poor general mental health. Therefore, we consider it meaningful to evaluate these relationships in patients with poor general mental health. Evaluating these relationships in patients with poor general mental health rather than pain conditions holds the potential to broaden the horizon of understanding the association between pain and mental illnesses. Some assessment criteria for central sensitisation [1,2,3] include not only physical symptoms but also mental symptoms, necessitating the expansion of the study population to include not just physical but also mental disorders.

In addition to the relationship between CS, depression, and anxiety, previous studies have reported the possibility of a relationship between CS and functional somatic symptoms [23,24]. CS has also been studied in relation to chronic pain and persistent physical symptoms [9]. For patients with chronic pain and persistent physical symptoms, understanding the mechanism of CS may lead to a better understanding of how the symptoms persist [25,26]. Depression and anxiety constitute the most crucial group of mental symptoms, and physical symptoms are intricately linked to central sensitisation. However, there has been no study that simultaneously evaluates the relationship between central sensitisation and depression, anxiety, and physical symptoms.

Therefore, we investigated the relationships of CS with depression, anxiety, and somatic symptoms, not limited to pain-related diseases but targeting general mental health outpatients. We hypothesised that there is a clear positive relationship of CS with depression, anxiety, and somatic symptoms (CS and depression, CS and anxiety, CS and somatic symptoms) in general mental health outpatients.

## 2. Materials and Methods

### 2.1. Study Design and Ethical Considerations

This cross-sectional study was based on the STROBE (strengthening the reporting of observational studies in epidemiology) statement (Appendix A) [27,28] and was conducted at the outpatient service of the Department of Psychosomatic Medicine at Toho University Medical Center. This clinical trial was conducted following the most recent revision of the Declaration of Helsinki and received approval from the Ethics Committee of Toho University School of Medicine (registration number A22086). Since existing data were used, informed consent was conducted through an opt-out method, clearly stating on the research institution’s website the right to withdraw from the study.

### 2.2. Participants and Diagnosis of Diseases

All individuals involved were new patients attending the Toho University Medical Center Omori Hospital from 1 May 2022 to 30 November 2022, and completed a self-administered questionnaire. We did not select patients based on age or sex. However, we excluded patients whose levels of depression and anxiety were significantly affected by schizophrenia, delusional disorder, and severe physical and mental disorders. Multiple physicians diagnosed the patients using the *Diagnostic and Statistical Manual of Mental Disorders, 5th edition* [29].

### 2.3. Evaluation of Depression, Anxiety, Physical Symptoms, and CS

Depression and anxiety were evaluated using the seven-item Hospital Anxiety and Depression Scale (HADS) for hospital patients [30]. The scales for depression and anxiety are scored from 0 to 21 points, and the cut-off values for both depression and anxiety are 8 points or more [31]. The reliability and validity of the Japanese version of the scale have been confirmed [32]. Somatic symptoms were assessed using the Somatic Symptom Scale-8 (SSS8) [33]. The Japanese version of the SSS8 has been linguistically and psychologically validated, demonstrating internal consistency [34]. CS was evaluated using the Japanese version of the Central Sensitisation Inventory (CSI9). The short form of CS consists of nine symptoms related to the CS syndrome. Each symptom was evaluated on a 5-point scale: never, rarely, sometimes, often, and always. Higher scores indicated a higher tendency for CS. Previous studies have evaluated the validity and reliability of this scale [35,36].

### 2.4. Confounding Variables

In the relationships of CSI9 with HADS depression, HADS anxiety, and SSS8, the following characteristics were extracted as potential confounding variables: age, sex, educational history, smoking history [37,38], and alcohol drinking history [39]. The participants were classified into three categories based on their educational history: junior high school, high school, and university or higher. The patient’s smoking history was categorised into two groups: people who currently smoke and people who are not presently engaged in smoking. Alcohol consumption history was divided into two categories: regular drinking and non-regular drinking, which encompasses former drinking and non-drinking.

### 2.5. Sample Size

We gathered information from 438 new patients visiting Toho University Omori Medical Center Hospital for the first time. We did not conduct a formal sample size calculation for this study since it was exploratory in nature. We evaluated the relationship between depression, anxiety, somatic symptoms, and CS using logistic regression analysis. We assumed that there were five adjustment variables; therefore, a minimum of 50 people was required. After the overall analysis, we stratified the disease and conducted similar analyses. After excluding samples with insufficient data, the data of 415 patients were analysed. Disease-specific analyses were performed for depressive disorders (n = 103), somatic symptoms and related disorders (n = 84), adjustment disorders (n = 70), and anxiety disorders (n = 63), indicating an adequate sample size. Thus, we conclude that this analysis is logically acceptable.

### 2.6. Statistical Analysis

We descriptively evaluated the basic characteristics of the study population using main questionnaire scores (i.e., HADS anxiety, HADS depression, SSS8, and CSI9) and factors such as sex, age, and potential confounders (i.e., educational history, smoking history, and alcohol drinking history). Continuous variables are presented as mean (standard deviation [SD]) or median (interquartile range [IQR]) based on distribution.

After a descriptive evaluation of the characteristics of the study population, logistic regression analyses were used to assess the association of HADS anxiety, HADS depression, and SSS8 with CSI9 in the entire sample population, adjusting for age, sex, educational history, smoking history, and alcohol consumption. The CSI9 score was converted into binary variables for clinical assessment: scores below 20 or exactly 20 were labelled as 0, while scores exceeding 21 were labelled as 1 [36].

In our disease-specific analysis, we selected diseases such as depressive disorders (n = 103), somatic symptoms and related disorders (n = 84), adjustment disorders (n = 70), and anxiety disorders (n = 63) because each had a sample size of over 50. Both the unadjusted and adjusted odds ratios (AORs) showed statistical significance at *p* < 0.05 (two-tailed). The analyses were all conducted with STATA^®^ version 14.

## 3. Results

We utilized data from a total of 415 participants, with an average age of 42.3 years (standard deviation = 1.0 year), of whom 245 were female. Table 1 shows the basic characteristics, classification of diseases, and main questionnaire scores (i.e., HADS anxiety, HADS depression, SSS8 score, and CSI9) of the participants.

In order of frequency, participants’ disorders included depressive disorders (n = 103), somatic symptoms and related disorders (n = 84), adjustment disorders (n = 70), anxiety disorders (n = 63), sleep–wake disorders (n = 37), feeding and eating disorders (n = 30), and other diseases (n = 28). Multiple logistic regression analysis was performed on the total sample (Table 2) and revealed that clinical CS was significantly associated with a greater severity of HADS anxiety (AOR: 1.24, 95% confidence interval CI: 1.17–1.32, *p* < 0.01), HADS depression (AOR: 1.18, 95% CI: 1.12–1.24, *p* < 0.01), and SSS8 (AOR: 1.34, 95% CI: 1.26–1.42, *p* < 0.01), after controlling for potential confounders.

In our disease-specific analysis, the relationships of clinical CS with HADS anxiety, HADS depression, and SSS8 were evaluated. Among the anxiety disorders (Table 3), depressive disorders (Table 4), and somatic symptoms and related disorders (Table 5), the associations of HADS anxiety, HADS depression, and SSS8 with clinical CS were positive (*p* < 0.01 in all analyses).

Among adjustment disorders (Table 6), only the association between SSS8 and clinical CS was positive (AOR: 2.15, 95% CI: 1.40–3.30, *p* < 0.01).

The SSS8 tended to exhibit a stronger positive correlation with the CSI9 than with the HADS anxiety and depression.

## 4. Discussion

This cross-sectional study investigated the association of central sensitisation and depression, anxiety, and somatic symptoms, revealing significant positive associations among these factors. In our disease-specific analysis, the SSS8 exhibited a stronger positive correlation with clinical CS than with HADS depression or HADS anxiety. These results indicate that CS is closely related to depression, anxiety, and somatic symptoms and has a particularly strong positive association with somatic symptoms. Disease analysis showed that a positive relationship between SSS8 and clinical CS was generally observed, and this relationship was stronger than that between HADS depression and HADS anxiety in depressive and somatoform disorders.

Previous studies have reported positive associations between CS, depression, and anxiety in fibromyalgia [12,13], breast cancer [14], temporomandibular disorders [15], and chronic lower back pain [16]. Our study, like many previous studies, also recognises a positive correlation. We think this is because our study focused on patients with mental disorders, who have a high prevalence of depression and anxiety, and because our sample size was larger compared to clinical studies. The present findings suggest that this association extends not only to these physical conditions but also to general mental disorders. Furthermore, the positive relationship of CS with depression, anxiety, and somatic symptoms was thought to be the strongest for somatic symptoms. Therefore, it may be possible that embodied symptoms represented by the functional somatic syndrome (FSS) are related to CS [40,41]. The term functional somatic syndrome has been applied to several related syndromes characterised more by symptoms, suffering, and disability than by consistently demonstrable tissue abnormalities [42]. CS is a state in which symptoms are strongly perceived and closely related to FSS, where symptoms appear to the extent that daily life is affected. Previous studies have reported that functional somatic symptoms are most severe when affected by CS [23,24]. Physical symptoms arise from both peripheral and central processes, and previous research has demonstrated the importance of central symptom processing [24]. Central sensitisation is characterised by hypersensitivity in this processing. However, depression and anxiety are mental symptoms primarily originating from central processes. Hence, the association between central sensitisation and physical symptoms may be more significant than that with depression or anxiety.

Among anxiety disorders, depressive disorders, and somatic symptoms and related disorders, HADS depression, HADS anxiety, and SSS8 were associated with clinical CS. This association suggests a strong potential for the positive association of central sensitisation with anxiety, depression, and somatic symptoms in each respective mental disorder. Let us assess the strength of the associations in each disease through an analysis specific to each condition. Among depressive disorders, HADS anxiety, HADS depression, and SSS8 were positively associated with clinical CS. This relationship was stronger with SSS8 than with HADS anxiety and HADS depression. This stronger relationship suggests a particularly strong positive association between somatic symptoms and central sensitisation in depressive disorders. Among somatic symptoms and related disorders, HADS anxiety, HADS depression, and SSS8 were positively associated with CSI. This relationship was stronger with SSS8 than with HADS anxiety and HADS depression. This stronger relationship also suggests a particularly strong positive association between somatic symptoms and central sensitisation in somatic symptoms and related disorders. Among anxiety disorders, HADS anxiety, HADS depression, and SSS8 were positively associated with clinical CS. This relationship was stronger with HADS anxiety than with HADS depression and SSS8. This stronger relationship suggests a particularly strong positive association between anxiety and central sensitisation in the anxiety disorder group. The reason for the relatively low correlation between somatic symptoms and central sensitisation in anxiety disorders is unclear. Anxiety disorders exhibit diversity, which may have contributed to the weaker association [29]. However, by expanding the sample size and conducting a detailed analysis of individual disorders within the anxiety disorder group, the underlying causes may become clearer. Among adjustment disorders, only SSS8 was related to clinical CS, and no relationship was observed between HADS depression and HADS anxiety. This lack of a relationship may be because adjustment disorders, which are not core symptom-based diseases, have multiple vague symptoms [29].

This study suggests that the potential connection between clinical CS and depression, anxiety, and physical symptoms across a range of mental disorders is significant; however, it has several limitations. First, because of its cross-sectional design, this study cannot directly imply causality that high depression, anxiety, and somatic symptom scores lead to an increase in CS or vice versa; rather, it can only speculate a statistical association. A more robust study design (e.g., a prospective cohort study) is required to confirm these results. Second, although the study included a statistically significant number of participants, the data were obtained from a single facility. Most of them sought treatment at the psychosomatic department affiliated with the university hospital, which limited the representativeness of the sample and the external validity of this study. Therefore, studies from other facilities are necessary to generalise the results. A large-scale prospective cohort study involving participants from other facilities is necessary to clarify the results of this study further.

This study refers to the possibility of elucidating the causes of physical symptoms through CS measurement in patients with common mental disorders included in pain-related conditions. Physical symptoms, even in those with apparent organic diseases, contain a certain degree of psychological influence [23,24,25,26]. The measurement of CS in this study could confirm the extent of psychological influence and potentially contribute to treatment strategies. Evaluating the psychological impact included in physical illnesses from a public health perspective may reduce economic losses due to medically unexplained symptoms [43].

## 5. Conclusions

There were significant associations between anxiety, depression, somatic symptoms, and clinical CS in patients attending general mental health outpatient services. These findings highlight the importance of evaluating somatic symptoms when assessing CS in mental disorders. Assessing central sensitisation in patients can be useful in distinguishing whether their physical symptoms have a physical or central origin.

## Figures and Tables

**Table 1 life-14-00612-t001:** Basic characteristics of the participants (n = 415).

	Total
Women, n (%)	245 (59.0)
Age, median (IQR)	42 (25–57)
Educational history, n (%)	
Junior high school	31 (7.5)
High school	122 (29.4)
University and over	262 (63.1)
Smoking history, n (%)	
Current smoking	72 (17.3)
Not presently engaged in smoking	343 (82.7)
Drinking history, n (%)	
Regular drinking	135 (32.5)
Non-regular drinking	280 (67.5)
Diagnosis, n (%)	
Adjustment disorder	70 (16.9)
Anxiety disorders	63 (15.2)
Depressive disorders	103 (24.8)
Feeding and eating disorders	30 (7.2)
Sleep–wake disorders	37 (8.9)
Somatic symptoms and related disorders	84 (20.2)
Other diseases *	28 (6.8)
HADS anxiety, median (IQR)	10 (6–13)
HADS depression, median (IQR)	10 (6–14)
SSS-8, median (IQR)	12 (8–18)
CSI score, mean (SD)	16.4 (7.9)
Clinical high CSI score 21 and over, n (%)	116 (28.0)

* Other diseases include bipolar disorder (n = 5), substance-related and addictive disorders (n = 4), neurodevelopmental disorders (n = 9), dissociative disorders (n = 1), personality disorders (n = 3), and Schizophrenia spectrum disorders (n = 6). IQR means interquartile range. SD means standard deviation.

**Table 2 life-14-00612-t002:** Association of HADS anxiety, HADS depression, and SSS8 with clinical CSI of the total sample in a cross-sectional study in Japan, 2022 (n = 514).

	Clinical CSI					
	Model 0 (Not Adjusted)		Model 1 (Adjusted)		Model 2 (Adjusted)	
	OR (95% CI)	*p* Value	AOR (95% CI)	*p* Value	AOR (95% CI)	*p* Value
HADS Anxiety	1.24 (1.17–1.31)	<0.01	1.24 (1.17–1.32)	<0.01	1.24 (1.17–1.32)	<0.01
Age			1.00 (0.99–1.00)	0.95	1.08 (0.99–1.01)	0.84
Sex			0.60 (0.37–0.99)	0.04	0.51 (0.30–0.85)	0.01
Educational history					0.98 (0.67–1.44)	0.94
Smoking history					2.14 (1.17–3.92)	0.01
Drinking history					1.18 (0.71–1.98)	0.52
HADS Depression	1.17 (1.12–1.23)	<0.01	1.18 (1.13–1.24)	<0.01	1.18 (1.12–1.24)	<0.01
Age			1.00 (0.99–1.01)	0.87	1.00 (0.99–1.01)	0.74
Sex			0.56 (0.34–0.90)	0.02	0.47 (0.28–0.79)	<0.01
Educational history					0.97 (0.66–1.41)	0.86
Smoking history					2.27 (1.25–4.13)	<0.01
Drinking history					1.10 (0.66–1.84)	0.71
SSS8	1.33 (1.26–1.41)	<0.01	1.34 (1.26–1.42)	<0.01	1.34 (1.26–1.42)	<0.01
Age			1.01 (1.00–1.03)	0.16	1.01 (0.99–1.02)	0.23
Sex			0.67 (0.38–1.17)	0.16	0.55 (0.30–1.00)	0.05
Educational history					0.98 (0.62–1.55)	0.94
Smoking history					2.12 (1.07–4.23)	0.03
Drinking history					1.36 (0.75–2.48)	0.31

HADS means Hospital Anxiety and Depression Scale. SSS8 means Somatic Symptom Scale-8. Model 0: not adjusted analysis. Model 1: adjusted by age and sex. Model 2: adjusted by age, sex, educational history, smoking history, and drinking history. In terms of clinical assessment, CSI9 scores were converted into binary variables; scores below 20 were categorized as 0, whereas scores of 21 and higher were categorized as 1.

**Table 3 life-14-00612-t003:** Association of HADS anxiety, HADS depression, and SSS8 with clinical CSI among anxiety disorders of a cross-sectional study in Japan, 2022 (n = 63).

	Clinical CSI					
	Model 0 (Not Adjusted)		Model 1 (Adjusted)		Model 2 (Adjusted)	
	OR (95% CI)	*p* Value	AOR (95% CI)	*p* Value	AOR (95% CI)	*p* Value
HADS Anxiety	1.30 (1.07–1.58)	<0.01	1.31 (1.07–1.60)	<0.01	1.51(1.15–1.99)	<0.01
Age			0.99 (0.96–1.02)	0.54	0.99 (0.95–1.03)	0.51
Sex			0.58 (0.12–2.78)	0.50	0.79 (0.15–4.26)	0.79
Educational history					0.18 (0.04–0.72)	0.02
Smoking history					1.31 (0.03–3.28)	0.33
Drinking history					1.39 (0.19–10.44)	0.75
HADS Depression	1.23 (1.07–1.41)	<0.01	1.23 (1.07–1.41)	<0.01	1.28 (1.10–1.52)	<0.01
Age			1.00 (0.97–1.03)	0.86	1.00 (0.96–1.03)	0.74
Sex			0.66 (0.13–3.25)	0.61	0.80 (0.15–4.13)	0.79
Educational history					0.27 (0.07–1.00)	0.05
Smoking history					0.87 (0.11–6.87)	0.90
Drinking history					0.92 (0.13–6.51)	0.93
SSS8	1.33 (1.14–1.56)	<0.01	1.33 (1.13–1.57)	<0.01	1.41 (1.15–1.72)	<0.01
Age			1.00 (0.96–1.04)	0.93	1.00 (0.96–1.04)	0.94
Sex			0.56 (0.10–3.08)	0.51	0.62 (0.10–3.94)	0.61
Educational history					0.30 (0.08–1.10)	0.07
Smoking history					0.28 (0.02–3.88)	0.34
Drinking history					1.65 (0.18–15.03)	0.34

HADS means Hospital Anxiety and Depression Scale. SSS8 means Somatic Symptom Scale-8. Model 0: not adjusted analysis. Model 1: adjusted by age and sex. Model 2: adjusted by age, sex, educational history, smoking history, and drinking history. In terms of clinical assessment, CSI9 scores were converted into binary variables; scores below 20 were categorized as 0, whereas scores of 21 and higher were categorized as 1.

**Table 4 life-14-00612-t004:** Association of HADS anxiety, HADS depression, and SSS8 with clinical CSI among depressive disorders of a cross-sectional study in Japan, 2022 (n = 103).

	Clinical CSI					
	Model 0 (Not Adjusted)		Model 1 (Adjusted)		Model 2 (Adjusted)	
	OR (95% CI)	*p* Value	AOR (95% CI)	*p* Value	AOR (95% CI)	*p* Value
HADS Anxiety	1.28 (1.14–1.43)	<0.01	1.28 (1.14–1.44)	<0.01	1.30 (1.16–1.47)	<0.01
Age			1.00 (0.98–1.03)	0.92	1.00 (0.98–1.03)	0.74
Sex			0.57 (0.22–1.48)	0.25	0.35 (0.12–1.04)	0.06
Educational history					1.57 (0.66–3.70)	0.31
Smoking history					3.16 (0.96–10.48)	0.06
Drinking history					1.46 (0.53–4.04)	0.47
HADS Depression	1.28 (1.14–1.43)	<0.01	1.22 (1.10–1.35)	<0.01	1.23 (1.11–1.37)	<0.01
Age			1.00 (0.98–1.03)	0.87	1.01 (0.98–1.03)	0.72
Sex			0.51 (0.98–1.03)	0.15	0.32 (0.11–0.91)	0.03
Educational history					1.43 (0.63–3.25)	0.40
Smoking history					3.64 (1.11–11.97)	0.03
Drinking history					1.07 (0.40–2.87)	0.90
SSS8	1.33 (1.18–1.49)	<0.01	1.36 (1.21–1.54)	<0.01	1.37 (1.21–1.56)	<0.01
Age			1.02 (0.99–1.05)	0.27	1.03 (0.99–1.06)	0.15
Sex			0.39 (0.13–1.19)	0.10	0.27 (0.08–0.93)	0.04
Educational history					1.95 (0.69–5.48)	0.21
Smoking history					1.91 (0.52–7.00)	0.33
Drinking history					0.96 (0.30–3.11)	0.94

HADS means Hospital Anxiety and Depression Scale. SSS8 means Somatic Symptom Scale-8. Model 0: not adjusted analysis. Model 1: adjusted by age and sex. Model 2: adjusted by age, sex, educational history, smoking history, and drinking history. In terms of clinical assessment, CSI9 scores were converted into binary variables; scores below 20 were categorized as 0, whereas scores of 21 and higher were categorized as 1.

**Table 5 life-14-00612-t005:** Association of HADS anxiety, HADS depression, and SSS8 with clinical CSI among somatic symptoms and related disorders of a cross-sectional study in Japan, 2022 (n = 84).

	Clinical CSI					
	Model 0 (Not Adjusted)		Model 1 (Adjusted)		Model 2 (Adjusted)	
	OR (95% CI)	*p* Value	AOR (95% CI)	*p* Value	AOR (95% CI)	*p* Value
HADS Anxiety	1.24 (1.10–1.41)	<0.01	1.24 (1.09–1.41)	<0.01	1.24 (1.09–1.41)	<0.01
Age			1.00 (0.97–1.03)	0.99	1.00 (0.97–1.03)	0.95
Sex			0.47 (0.13–1.71)	0.25	0.61 (0.16–2.33)	0.47
Educational history					0.94 (0.38–2.35)	0.90
Smoking history					1.00 omitted	
Drinking history					1.08 (0.33–3.57)	0.90
HADS Depression	1.16 (1.06–1.29)	<0.01	1.19 (1.06–1.34)	<0.01	1.18 (1.05–1.33)	<0.01
Age			0.99 (0.96–1.02)	0.39	0.99 (0.96–1.02)	0.40
Sex			0.36 (0.10–1.27)	0.11	0.42 (0.11–1.53)	0.19
Educational history					1.03 (0.42–2.52)	0.95
Smoking history					1.00 omitted	
Drinking history					1.27 (0.39–4.11)	0.69
SSS8	1.33 (1.16–1.51)	<0.01	1.32 (1.16–1.50)	<0.01	1.34 (1.16–1.54)	<0.01
Age			1.00 (0.96–1.03)	0.80	0.99 (0.96–1.03)	0.72
Sex			0.48 (0.10–2.26)	0.36	0.62 (0.12–3.10)	0.56
Educational history					0.52 (0.17–1.57)	0.25
Smoking history					1.00 omitted	
Drinking history					1.51 (0.38–6.03)	0.56

HADS means Hospital Anxiety and Depression Scale. SSS8 means Somatic Symptom Scale-8. Model 0: not adjusted analysis. Model 1: adjusted by age and sex. Model 2: adjusted by age, sex, educational history, smoking history, and drinking history. In terms of clinical assessment, CSI9 scores were converted into binary variables; scores below 20 were categorized as 0, whereas scores of 21 and higher were categorized as 1.

**Table 6 life-14-00612-t006:** Association of HADS anxiety, HADS depression, and SSS8 with clinical CSI among adjustment disorders of a cross-sectional study in Japan, 2022 (n = 70).

	Clinical CSI					
	Model 0 (Not Adjusted)		Model 1 (Adjusted)		Model 2 (Adjusted)	
	OR (95% CI)	*p* Value	AOR (95% CI)	*p* Value	AOR (95% CI)	*p* Value
HADS Anxiety	1.08 (0.95–1.23)	0.25	1.08 (0.94–1.23)	0.29	1.10(0.94–1.27)	0.23
Age			1.00 (0.97–1.03)	0.99	1.00 (0.96–1.03)	0.84
Sex			0.55 (0.18–1.67)	0.29	0.38 (0.12–1.26)	0.12
Educational history					3.02 (0.89–10.25)	0.08
Smoking history					1.72 (0.44–6.70)	0.44
Drinking history					1.02 (0.30–3.42)	0.98
HADS Depression	1.08 (0.96–1.21)	0.19	1.09 (0.96–1.23)	0.17	1.09 (0.95–1.24)	0.22
Age			1.00 (0.97–1.03)	0.97	1.00 (0.96–1.24)	0.86
Sex			0.49 (0.16–1.52)	0.22	0.34 (0.10–1.16)	0.09
Educational history					2.77 (0.85–9.09)	0.09
Smoking history					1.57 (0.40–6.10)	0.51
Drinking history					1.12 (0.34–3.67)	0.85
SSS8	1.69 (1.31–2.19)	<0.01	1.93 (1.35–2.75)	<0.01	2.15 (1.40–3.30)	<0.01
Age			1.07 (0.99–1.14)	0.07	1.06 (0.98–1.14)	0.12
Sex			1.50 (0.22–10.46)	0.68	0.68 (0.08–5.49)	0.72
Educational history					6.45 (0.96–43.19)	0.06
Smoking history					4.05 (0.46–35.69)	0.21
Drinking history					2.36 (0.36–15.50)	0.37

HADS means Hospital Anxiety and Depression Scale. SSS8 means Somatic Symptom Scale-8. Model 0: not adjusted analysis. Model 1: adjusted by age and sex. Model 2: adjusted by age, sex, educational history, smoking history, and drinking history. In terms of clinical assessment, CSI9 scores were converted into binary variables; scores below 20 were categorized as 0, whereas scores of 21 and higher were categorized as 1.

## Data Availability

The datasets used and analysed during this study are available from the corresponding author upon reasonable request.

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
