# Peer review of "The Association of Central Sensitisation with Depression, Anxiety, and Somatic Symptoms: A Cross-Sectional Study of a Mental Health Outpatient Clinic in Japan"

_life, 2024, doi:10.3390/life14050612_

Round 1

Reviewer 1 Report

Comments and Suggestions for Authors

This article provides new insights into the role of central sensitization in various diseases, contributing valuable information for future research and clinical practice. However, there are several details that could be further refined:

Introduction:

- The article points out a current research gap – the relationship between central sensitization and mental health issues has been less studied. It is suggested that the authors further emphasize the importance of this research gap, elucidating the significance of filling this void for understanding central sensitization and related mental health issues. Additionally, while previous studies focused on participants "limited to specific pain-related diseases," this study targeted "general mental health patients." It is recommended to supplement an explanation of the innovation and research significance of this approach.

- It is suggested to add supplementary research background to reflect the importance of studying central sensitization in relation to mental health issues.

- The hypotheses are not sufficiently specific. It is recommended that the authors clarify whether the relationship between CS and depression, anxiety, and somatic symptoms is hypothesized to be positive or negative, and further explain why such hypotheses are proposed, whether there is prior theoretical or research support for these hypotheses, to strengthen the rationale and persuasiveness of the hypotheses.

Materials and Methods:

- While exclusion criteria are mentioned, they are not detailed. It is advised that the authors provide a detailed description of both inclusion and exclusion criteria to reflect the representativeness of the sample under evaluation.

- The authors provide a detailed list of tools used to assess depression, anxiety, somatic symptoms, and central sensitization, but do not explain these measurement tools. It is suggested that the authors further introduce the rationale for the selection of these tools, their psychometric properties, and why they are suitable for the target population of this study.

Discussion:

- The authors should delve deeper into why the association between central sensitization and somatic symptoms is more significant than that with depression or anxiety. Furthermore, regarding the differences in central sensitization manifested in different mental illnesses, the authors should provide possible explanations or cite relevant studies to enhance the depth and breadth of the discussion.

- The discussion section should include a critical analysis of the limitations of the study. For example, the authors should discuss limitations such as the cross-sectional study design and potential biases introduced by the sample source. Additionally, provide directions for future research, including addressing the current study's limitations and further exploring possible avenues for investigating the relationship between central sensitization and various mental illnesses.

Conclusion:

- The authors mention the importance of assessing central sensitization in clinical practice. It is recommended that the authors further elaborate on how these findings specifically impact clinical practice, and how physicians and healthcare professionals can utilize this information to improve patient management and treatment.

Author Response

Dear Editors and Reviewers

Thank you for reviewing our manuscript. We have received your kind letter relating to the publication of our manuscript, “Association of central sensitisation with depression, anxiety, and somatic symptoms: a cross-sectional study of a mental health outpatient clinic in Japan” in life. We have revised our manuscript following the reviews. Italic parts are the reviewers’ comments and the following underlined sections detail our revisions, The altered parts are also underlined in the revised manuscript.

Reviewer 2 Report

Comments and Suggestions for Authors

Dear Authors kindly highlight the positive or negative association between CS & depression and anxiety.

1. You hypothesized that there is a clear relationship of CS with depression, anxiety, and somatic symptoms in general mental health outpatients is confusing.

2. Kindly attach the STROBE checklist in the supplementary section.

3. Is there data related to non-smokers, please attach and apply stats.

4. Please share the duration of illness for these categories e.g. depression, anxiety, and adjustment disorders. How it's possible to make arbitrary conclusions that CS is significantly affected in an individual with the first episode or prolonged history? How to conclude results based on such biases? 

5. In our disease-specific analysis, the SSS8 exhibited a stronger correlation with clinical CS than with HADS depression or HADS anxiety- this sentence is unclear. Kindly highlight and explain stronger correlations.

6. Please rewrite the discussion based on your findings and clinically meaningful outcomes.

Comments on the Quality of English Language

Needs to be scientifically clear and concise.

Author Response

(The authors gave the same response as above.)

Reviewer 3 Report

Comments and Suggestions for Authors

The topic of the study is very interesting and the introduction well written and well grounded. The objectives of the study are clearly presented.

The abstract is correct. It is adequate to the issues presented in the article.

The part concerning the research procedure is compatible with the requirements.

Some recommendations for improving the article could be:

- The mention of ethics committee approval and adherence to the Helsinki Declaration and f the Psychosomatic Medicine Department at Toho University Hospital is good. However, consider detailing the process of obtaining informed consent and how participants were informed about their rights, including confidentiality and voluntary participation.

- Please provide more detailed information on the internal consistency testing of the instruments used in the study, the method used and the indicators obtained. Were these tools previously validated in the context of Japan?

- Also, the inclusion of examples of instrument items might be appropriate

- The discussion part can be started by presenting the objective of the study and then mentioning the results obtained.

- You've done well in comparing your findings with existing studies. Continue this by providing more context on how your results align or differ from previous research and the possible reasons for these similarities or differences.

- A broad discussion of the clinical and psychotherapeutic implications of these results, with reference to the results of published studies, is absolutely necessary.

- Provide specific recommendations for clinical practice based on your findings.

- Provide more specific policy recommendations based on your findings. How can healthcare systems or policymakers use this information to improve care?

- Also, future research directions concerning the relationships between variables could be mentioned.

Author Response

(The authors gave the same response as above.)

Round 2

Reviewer 1 Report

Comments and Suggestions for Authors

The quality of the revised manuscript has improved a lot. The authors responded to all my concerns.

Author Response

Dear Editors and Reviewers

Thank you for reviewing our manuscript. We have received your kind letter relating to the publication of our manuscript, “Association of central sensitisation with depression, anxiety, and somatic symptoms: a cross-sectional study of a mental health outpatient clinic in Japan” in life.

REVIEWER 1 COMMENTS

The quality of the revised manuscript has improved a lot. The authors responded to all my concerns.

Response: Thanks for your kind comments.

Reviewer 2 Report

Comments and Suggestions for Authors

The authors might want to submit the pending files before the next review.

Comments on the Quality of English Language

The authors need more work on the flow of the manuscript.

Author Response

(The authors gave the same response as above.)
